# Eight-Year Retrospective Study of Young Adults in a Diabetes Transition Clinic

**DOI:** 10.3390/ijerph182312667

**Published:** 2021-12-01

**Authors:** Aarooran Sritharan, Uchechukwu L. Osuagwu, Manjula Ratnaweera, David Simmons

**Affiliations:** 1Diabetes, Obesity and Metabolism Translational Research Unit, School of Medicine, Western Sydney University, Campbelltown, NSW 2560, Australia; aaroorans@gmail.com (A.S.); l.osuagwu@westernsydney.edu.au (U.L.O.); 2Translational Health Research Unit (THRI), School of Medicine, Western Sydney University, Campbelltown, NSW 2560, Australia; 3Waikato Hospital, Hamilton 3240, New Zealand; Manjula.Ratnaweera@waikatodhb.health.nz; 4Macarthur Diabetes Endocrinology and Metabolism Service, Campbelltown Hospital, Campbelltown, NSW 2560, Australia

**Keywords:** diabetic ketoacidosis, mental health, type 1 diabetes, transition, glycaemia

## Abstract

The transition of people from paediatric to adult diabetes services is associated with worsening glycaemia and increased diabetes-related hospitalisation. This study compared the clinical characteristics of those with and without mental health conditions among attenders at a diabetes young adult clinic diabetes before and after changes in service delivery. Retrospective audit of 200 people with diabetes attending a Sydney public hospital over eight years corresponding to the period before (2012–2016) and after (2017–2018) restructuring of a clinic for young adults aged 16–25 years. Characteristics of those with and without mental health conditions (depression, anxiety, diabetes related distress, eating disorders), were compared. Among clinic attenders (type 1 diabetes *n* = 184, 83.2%), 40.5% (*n* = 89) had a mental health condition particularly, depression (*n* = 57, 64%), which was higher among Indigenous than non-Indigenous people (5.6% vs. 0.8%, *p* = 0.031) but similar between diabetes type. Over eight years, those with, compared with those without a mental health condition had higher haemoglobin A1c (HbA1c) at the last visit (9.4% (79 mmol/mol) vs. 8.7% (71 mmol/mol), *p* = 0.027), the proportion with diabetic ketoacidosis (DKA 60.7% vs. 42.7%, *p* = 0.009), smoking (38.4 vs. 13.6%, *p* = 0.009), retinopathy (9.0 vs. 2.3%, *p* = 0.025), multiple DKAs (28.4 vs. 16.0%, *p* = 0.031) were significantly higher. Having a mental health condition was associated with 2.02 (95% confidence intervals 1.1–3.7) fold increased risk of HbA1c ≥9.0% (75 mmol/mol). Changes to the clinic were not associated with improvements in mental health condition (39.0% vs. 32.4%, *p* = 0.096). In conclusion, we found that mental health conditions, particularly depression, are common in this population and are associated with diabetes complications. Diabetes type and clinic changes did not affect the reported mental health conditions. Additional strategies including having an in-house psychologist are required to reduce complication risks among those with mental health conditions.

## 1. Introduction

The transition of people from paediatric to adult diabetes services is associated with worsening glycaemia and increased diabetes-related hospitalisation [1]. This transition period correlates with increased risk-seeking behaviour including drugs, alcohol and unsafe sex [1]. The transition period generally corresponds to people moving out of home alongside changes in social and employment situations, and is associated with emotional and financial hardship. Under such circumstances, several people lose continuity of professional care, which may affect outcomes [2]. Poor glycaemic control and mental health conditions, such as depression, anxiety, eating disorders and diabetes distress, are common in this population [3,4]. However, diagnoses of mental health conditions are often missed, misdiagnosed or poorly managed exacerbating the poorer prognosis of affected young people [5,6].

The increased prevalence of diabetes in adolescence and young adulthood around the world warrants effective treatment models and guidelines for healthcare professionals involved in their care [2]. Implementing a well-structured program for these transitioning young adults is pivotal for long-term outcomes and continued compliance with diabetes self-management [4,7]. Such programs should include education for paediatric care providers, pre-pregnancy programs for women [8], and screening for mental health co-morbidities [3], thus optimising transition of care from paediatric to adult clinic. The transition conversations should happen early with the care appropriately tailored to the developmental level and unique needs of emerging adults [9]. These strategies may improve glycaemia, reduce the frequency of diabetic ketoacidosis (DKA)-related hospitalisations, and improve clinic attendance [4,10], thereby reducing short term personal and financial burden [11]. Maintaining lower haemoglobin A1c (HbA1c) in people over a long period of time reduces complications and improves quality of life with added economic benefits [12]. This study compared glycaemia and hospitalisation risks of people attending a diabetes transition clinic with and without mental health conditions, as well as the effects of clinic structural changes on the outcomes.

## 2. Subjects and Methods

A tertiary public hospital located in Sydney, one of the few major city hospitals with a transition diabetes service situated in the same facility as the paediatric diabetes service. The hospital serves 3 local government areas with an estimated population of 283,743 residents in 2016, including 38,622 (13.6%) aged 15–24 years [13] of whom approximately 400 were known to be living with diabetes [14].

### 2.1. The Diabetes Transition Clinic

The diabetes transition clinic is a multidisciplinary (endocrinologist, educator, dietitian) clinic that commenced in 2011 with one of the educators serving as a coordinator [15]. Diabetes educators are clinicians who specialise in the provision of diabetes self-management education for people with diabetes. People were considered to have a mental health condition if there had been a formal diagnosis by either a psychiatrist, psychologist or general practitioner and/or following an assessment by the diabetes educator or endocrinologist. Mental health support was provided through the youth health service or through communication with the general practitioner for referral to a private psychologist in the community.

An audit of our clinic conducted in 2016 [6] showed high rates of mental health comorbidities (59%), DKA-related hospitalisations since diagnosis (39%) and prior pregnancies (23%). These prompted a re-assessment of the clinic’s mode of operation in the latter half of 2016, including the introduction of an afternoon clinic operating from 3.00–8.00 p.m. (previously a Monday morning clinic) to optimise clinic attendances of those with strict working or studying schedules, a second diabetes educator, the use of structured clinic sheets for documentation of consultations and greater links with youth health services. Continuous glucose monitoring systems (CGMS) became more accessible for selected young adults (up to 21 years) through a Federal Government programme. A separate lounge with a television system was provided for entertainment and display of information.

The target levels for optimal control of the clinical measures were set as: HbA1c ≤ 7.0% for type 1 diabetes and ≤6.5% for type 2 diabetes, systolic blood pressure (sBP) < 130 mm Hg, diastolic blood pressure (dBP) < 80 mm Hg, total cholesterol ≤5.0 mmol/L, total triglycerides (TAGs) < 2.0 mmol/L, high density lipoprotein (HDL) cholesterol > 1.0 mmol/L, and low-density lipoprotein of LDL < 2.5 mmol/L [16].

### 2.2. Data Collection

The South Western Sydney Local Health Research Ethics Committee (LNR/16/LPOOL/180) approved the study. People were included if they were referred to the transition clinic (aged 16–25 years) between 2012 and 2019. Patient demographics, sources of referrals, metabolic, attendance, reasons for leaving the clinic and clinical history data were collected. Nephropathy included microalbuminuria (albumin/creatinine ratio (ACR) ≥ 2.5 mg/mmol for males and ≥3.5 mg/mmol for females), retinopathy was confirmed by an optometrist or ophthalmologist, and neuropathy by a clinical assessment. Glycaemia was measured either using a point-of-care HbA1c machine (Bayer DCA 2000 Blood Analyser, NSW, Australia), at the hospital laboratory (HPLC on Bio-Rad Variant II methodology), or through external pathology providers. Change in HbA1c over time was calculated as last HbA1c within the year minus first HbA1c. Driving requirement compliance refers to the recommendation, which requires the people to check their blood glucose level regularly before driving, carry quick acting carbohydrates and ensure that the reading is ≥5 mmol/L to avoid hypoglycaemia while driving.

### 2.3. Statistical Analysis

SPSS Version 26 (IBM Australia, St Leonards, NSW, Australia) was used for data analysis. One-way ANOVA was used to compare continuous variables and Chi-square test for proportions. An independent samples median test of the HbA1c at the end of the second year in clinic was conducted to compare the change in HbA1c from the baseline (commencement of clinic), before and after changes to the clinic were implemented. Binomial logistic regression was conducted to assess the impact of mental health on glycaemia, having one or more DKA events, and other diabetes complications. Age and diabetes duration were taken as continuous variables. The binomial regression was repeated, without HbA1c (model 2) to assess the impact of mental health conditions on DKA risk with the existing degree of hyperglycaemia. Indigenous status was excluded in the logistic regression due to small numbers. All tests were two tailed and *p* < 0.05 were considered significant. The calculated standardized mortality ratio (SMR: Observed/expected number of deaths (using Australian Bureau of Statistics [ABS] data) × 100) under or over 100 indicates lower or higher mortality compared with the general population in the same age group (0.4/1000 in the 20–25-year age group).

## 3. Results

### 3.1. Characteristics

Figure 1 shows that of the 260 referred people, 66% attended via the local paediatric diabetes clinic. The mean age at first assessment was 19.2 ± 1.8 years (range, 16–24 years), median attendance at the transition clinic was 1.0 (range, 7.9) years and had between 1–15 attendances (average of 4 times), over the study period as shown in Table 1.

Overall, 11.5% never attended the clinic, 7% were lost to follow up after one clinic visit, and 20% transferred to another service, including thirty-three patients aged below 25 years (15%) at the time of transfer (Figure 1). The referral rate increased in 2018–2019. Three of the 260 people who were referred to the clinic died between 2012–2019 (11.5 deaths (95%CI:.6 to 24.7) per 1000 people over 8 years) equivalent to an annual death rate of 1.4 per 1000 and a standardizsed mortality ratio of 350. Overall, 83.6% had type 1 diabetes, 14.1% type 2 diabetes and the rest (2.3%) had monogenic diabetes (MODY1, MODY3 or MODY4, *n* = 5).

Table 2 presents the clinical profile of the young adults, their treatment modalities and the diabetes-related complications reported in the transition clinic during the study period. Participants were mostly overweight, about one-third of them had hypertension and 21.1% had experienced multiple episodes of DKA over the study period. Insulin only treatment was predominant in this population and there was a reasonable proportion (*n* = 18, 50.0%) that were on Oral/GLP1 treatment only.

### 3.2. Comparison of People with and without Mental Health Conditions

In this transition clinic, mental health conditions were common (*n* = 89, 40.5% with more than one often being present) including depression (57, 64% of those with a mental health condition) anxiety (29, 33%), diabetes related distress (20, 22%), and eating disorders (4, 4.5%). There were six Aboriginal people in this study, five of whom (83.3%) had a mental health condition.

Table 3 compares the characteristics of people with and without a mental health condition. Despite having similar duration of diabetes, people with a mental health condition had a significantly higher HbA1c at their last clinic appointment, higher episodes of DKA-related hospitalisations, recurrent hospitalisations associated with DKAs and more long-term complications compared with those without a mental health condition. Smoking was over twice as common in people with a mental health condition. Between 2012–2017, three documented referrals to the youth health service occurred, with a further five referrals in 2018–2019. Only one encounter with a private psychologist was documented. The use of CGM was comparable between those with and without mental health condition (57.6% vs. 65.1%, *p* = 0.359) with similar rate of pregnancies (27.3% vs. 16.7%, *p* = 0.207). The proportion of patients who achieved target metabolic measures were largely comparable across metabolic measures.

Table 4 presents the adjusted odds ratio and 95% confidence intervals of factors associated with mental health condition in this study population. Having an HbA1c > 9% (75 mmol/mol) was the only factor associated with mental health conditions after adjusting for other potential confounders (adjusted odds ratio 2.02, 95% confidence intervals CI 1.10, 3.70). The relationship between mental health conditions and any DKA episode became significant after HbA1c was removed from the model (adjusted odds ratio 2.15, 95% CI 1.13, 4.07).

### 3.3. Glycaemic (HbA1c) Trend in the Clinic before and after Changes in Service Delivery

The mean HbA1c (9.2 ± 2.3%/77 ± 25 versus 8.8 ± 2.2%/73 ± 24 mmol/mol; *p* = 0.164) and rate of DKA (23.0%, *n* = 35 versus 14.8%, *n* = 21; *p* = 0.072) were similar before and after changes to the clinic. Figure 2 shows the trend in HbA1c across the years people attended the clinic compared to the baseline values Compared to baseline glycaemia, there were significant reductions in HbA1c each consecutive year the people attended the clinic with a mean reduction of 0.7% (8 mmol/mol) (*p* < 0.001) and 2.1% (23 mmol/mol, *p* = 0.016) from the baseline entry HbA1c by the end of years one and seven, respectively. The proportion using CGM also increased before and after changes to the clinic (9.7% versus 59.9%, *p* < 0.001).

## 4. Discussion

We have shown that mental health conditions were common in this population. Those with mental health conditions were more likely to have worse outcomes including poorer glycaemic control, more episodes of DKA, more long-term complications and being more likely to smoker. The proportion with a mental health condition was unaffected by the diabetes type. The mortality rate in this clinic population was high with a standardized mortality rate of 350. Although glycaemia improved substantially with the number of years young adults attended the clinic, there were no significant changes in HbA1c, and a non-significant reduction in DKA, following the changes to the clinic.

Mental health conditions were common in this cohort with higher rates than that reported in the ABS National Survey of Mental Health and Wellbeing [17], with respect to any mental health disorder (26.4 versus 40.5% in this study), depressive disorder (5.4 versus 33%) and anxiety disorder (15.4 versus 64%) among similar age group (16–24-year-old). Psychosocial conditions although prevalent in transition clinics are often less frequently reported as demonstrated in a recent review of 18 studies, where only 17% reported on the psychosocial conditions in transition clinics [15]. Since well-structured diabetes transition programmes improve diabetes management and psychological interventions improve glycaemic control in children and adolescents with diabetes [15], the findings of this present study provides further evidence of the need to integrate mental health support as part of the transition clinic. The reason for the high mortality rate is unclear but is likely associated with the high DKA, severe hypoglycaemia rates and the high prevalence of mental health conditions. No documented case of suicide was identified.

Young adults with mental health conditions have worse diabetes outcomes driven by factors including lack of clinic attendance and non-compliance with medications and diet [3]. However, in this study, the number of clinic check ins (attendance) was similar between those with and without mental health condition but the findings of more hyperglycaemia and frequent diabetes related hospitalisations in people with mental health conditions are consistent with the wider literature [6,11]. Prior studies reported difficulties of achieving optimal management of diabetes among young people with mental health conditions [3] and those with diabetes are a high risk group [18]. The stigma of being labelled with a mental health condition and disclosing personal information to another stranger can be frightening to young adults and this complicates the management of diabetes [19]. The low documented rate of youth service/private psychologist review of participants in this study is an indication of communication gap between the clinic and external mental health services which may hinder people from receiving mental health intervention. Thus, having an onsite psychologist may enhance communication, early recognition and management to facilitate optimal diabetes management [3].

The patient’s ability to meet target metabolic measures was also affected by their mental health status but significant only for HDL target (Table 2). A past study showed a strong link between dyslipidaemia dysregulations and mental illness with the authors suggesting that the unhealthy lifestyle and a poor adherence to medical regimen, which are prevalent among psychiatric patients, may contribute to this finding in addition to the reported profound influence of specific psychotropic medications on lipid regulations. [20] In this study, the smoking rate was significantly higher among people with diabetes who had a mental health condition compared to those without and as shown in one study, the smoking rate may also increase alongside the severity of the mental health condition [21]. Our findings are in line with past studies that found higher smoking rates in people with mental health conditions compared with the general Australian population [22,23,24]. The studies also showed that people with mental health condition have higher levels of nicotine dependence [25], and a disproportionate health and financial burden from smoking [24] which inadvertently would increase the burden of diabetes care.

The changes in the transition clinic did not significantly improve glycaemic control in this cohort, which is similar to previous follow up studies [26], possibly due to an increase in the number of referrals to the clinic during the second audit and that none of the changes in the clinic addressed the mental health aspects of people. However, the transition interventions may have reduced the number of DKA events, as there were non-significantly fewer DKAs in the second audit period. Different models of care have been proposed for improving outcomes of young people with diabetes. Some models involve care delivery by a combination of a transition clinic with a coordinator while others involve either just a clinic or coordinator [9,15]. In a meta-analysis, programs that included both a transition coordinator and a dedicated transition clinic were more effective at maintaining glycaemic control and/or preventing its worsening during transition compared to those that involve either component alone, but it was difficult to ascertain which element of the transition programs was most effective at improving diabetes outcomes [15].

### Limitations and Strengths

The study was retrospective with no control group making it difficult to differentiate between the improvements that were related to patient factors from those that were primarily from the intervention. Numbers and period after the change in the clinic were limited. The people were from a single hospital so our results may not apply to other patient groups in regions. The changes were multifaceted, making it difficult to differentiate the effect of any one intervention. We did not enquire about substance use and their effects on DKA and HbA1c, which were shown previously [26] were unaccounted for in our analysis. The retinal screening was performed outside of the hospital and many had not had it performed, which may explain the lack of difference in retinopathy screening. The mental health diagnoses were often not formal due to limited access to mental health services. Routine questionnaire can be administered while the people are waiting. In addition, since this was an audit, we did not provide data on the use of substances such as illicit drugs which was shown to be high in young people with (17.8%) [26] and without diabetes (12.7%) [17] in previous studies but not always documented in clinical data. Despite these limitations, the study has some strengths including the low default rate of people i.e., those that never attended the clinic after the initial referral and the large sample size (largest Australian study to date) compared with previous studies [11].

## 5. Conclusions

The high proportion with a mental health condition and limited access to psychological support were a significant problem strengthening the case for psychologist support into Australian diabetes transition services. Among young people with diabetes, those who smoke, with poor glycaemic control, experience more episodes of DKA and have long-term complications are more likely to have a mental health condition. However, changes to a well-structured multidisciplinary diabetes transition clinic including, an evening clinic and additional staffing did not significantly improve glycaemia with or without a mental health condition. Despite emerging strategies to help young adults with mental health issues, a diagnosis of a chronic disorder is an additional burden. Peer groups for affected individuals to interact with each other improves outcome of people with mental health condition [27]. Furthermore, having an in-house psychologist will initiate early intervention to improve patient satisfaction and optimal diabetes care. Strategies that involve the effective collaboration of primary care providers and mental health specialists/psychologists using principles of measurement-based stepped care and treatment to target the high-risk population identified in this study can substantially improve patients’ health and functioning while reducing overall health care costs.

## Figures and Tables

**Figure 1 ijerph-18-12667-f001:**
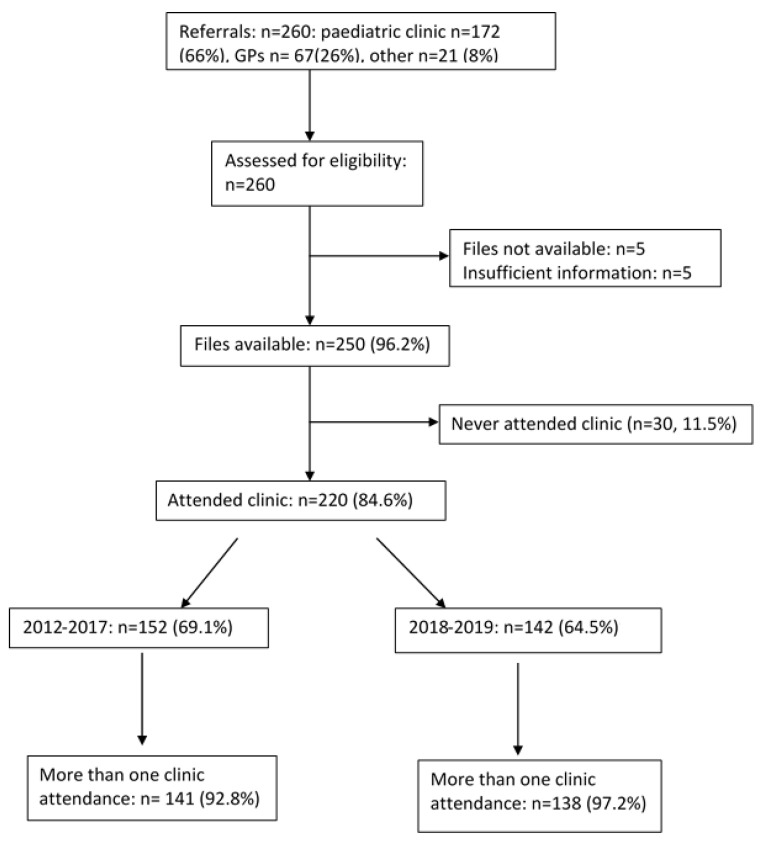
Flow of people in this study. The = % add up to more than 100% because 74 attended during both audits, 78 only in audit 1 and 68 only in audit 2.

**Figure 2 ijerph-18-12667-f002:**
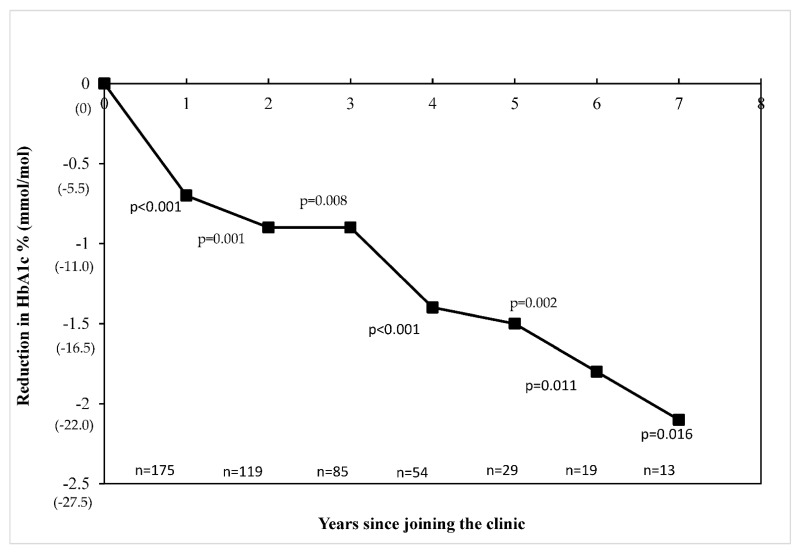
Change in heamoglobin A1c over the years attended clinic.

**Table 1 ijerph-18-12667-t001:** Characteristics of the sample population with diabetes in the transition clinic (2012–2018).

Variables	All
N (%)	220
Period	
2012–2016	78 (35.5)
2019–2019	68 (30.9)
2012–2019	74 (33.6)
Socio-demography	
Current age, mean (SD)	22.6 (3.1)
Age at first assessment, mean (SD)	18.6 (2.0)
Female, *n* (%)	92 (44.5)
Place of Birth—Australia	205 (93.2)
Aboriginal or Torres Strait Island	6 (2.7)
Smoking, *n* (%)	44 (20.0)
Alcohol, *n* (%)	108 (49.1)
Diabetes type	
Type 1 diabetes	184 (83.6)
Type 2 diabetes/Other	36(16.4)
Glycaemia, mean (SD)	
HbA1c at entry, % (mmol/mol)	9.9 ± 2.6 (84 ± 29)
Mean HbA1c	9.2 ± 2.0 (77 ± 22)
HbA1c at last visit, % (mmol/mol)	8.9 ± 2.3 (74 ± 25)
Reduction in HbA1c since first attendance, %(mmol/mol)	0.9 ± 2.8 (10.1 ± 29.8)
Driving requirement compliance, *n* (%)	89 (40.5)
Age of diagnosis (years), mean (SD)	12.2 ± 5.9
Diabetes duration (years), mean (SD)	9.8 ± 6.3
Number of clinic check ins, mean (SD)	6.1 ± 5.0
Number of years in clinic, mean (SD)	2.2 ± 1.7

**Table 2 ijerph-18-12667-t002:** Metabolic outcomes, treatment and complications of the sample population with diabetes in the transition clinic (2012–2018).

Variables	All
Metabolic outcomes, mean (standard deviation)	
Lipids (mmol/L)	
Mean total cholesterol	4.7 ± 1.1
Mean TAGs	1.8 ± 1.8
Mean HDL	1.4 ± 0.4
Mean LDL	2.7 ± 0.8
Blood pressures (mmHg)	
Mean of mean sBP	124 ± 11
Mean of mean dBP	75 ± 7
BMI (kg/m^2^)	28.0 ± 7.7
Treatment, *n* (%)	
Insulin only	169 (76.8)
Insulin and oral/GLP1	35 (15.9)
Oral/GLP1 only ^a^	18 (50.0)
Complications, *n* (%)	
Neuropathy	17 (7.7)
Nephropathy ^†^	34 (15.5)
Retinopathy	11 (5.0)
Any long-term complication, *n* (%)	34 (15.5)
Hypertension, *n* (%)	77 (35.0)
Mental health condition, *n* (%)	89 (40.5)
DKA ^†^, *n* (%)	110 (50.0)
Multiple DKA episodes, *n* (%)	46 (21.1)
Hypoglycaemia admissions, *n* (%)	24 (10.9)
Multiple hypo episodes, *n* (%)	6 (2.8)
Any long-term complication, *n* (%)	34 (15.5)
Hypertension, *n* (%)	77 (35.0)
Target achieved, *n* (%)	
sBP < 130 mm Hg	142 (71.4)
dBP < 80 mmHg	148 (74.4)
HbA1c **	23 (10.7)
Total cholesterol < 5.0 mmol/L	106 (66.7)
Total TAGs < 2.0 mmol/L	122 (76.7)
Total HDL > 1.0 mmol/L	116 (81.7)
Total LDL < 2.5 mmol/L	61 (43.9)

^a^ Metformin only in those with type 2 diabetes. Abbreviations: HbA1c = haemoglobin A1c; GLP1 = Glucagon-like peptide−1 Receptor agonist; DKA = diabetic ketoacidosis; TAG = triglycerides; HDL = high density lipoprotein; LDL = low density lipoprotein; sBP = systolic blood pressure; dBP = diastolic blood pressure; BMI = body mass index ^†^ > 2.5 mg/mmol for males and >3.5 mg/mmol for females. ** ≤7% (53 mmol/mol) for people with Type 1 diabetes and ≤6.5% (48 mmol/mol) for those with type 1 diabetes/Other types of diabetes. *p* values were Chi-square test for discrete variables and t-test for continuous variables.

**Table 3 ijerph-18-12667-t003:** Comparison of people with and without mental health conditions.

Variables	Mental Health Conditions (*n* = 89)	No Mental Health Conditions (*n* = 131)	*p* Value
Current age, mean (± SD) years	22.8 (2.8)	22.5 (3.2)	0.816
Age at first assessment, mean (± SD) years	18.7 (1.96)	18.6 (2.0)	0.816
Gender Male, *n* (%) Female, *n* (%)	45 (50.6) 44 (49.4)	77 (59.2) 53 (40.8)	0.205
Type of Diabetes 1, *n* (%) 2/other, *n* (%)	74 (83.1) 15 (16.9)	109 (83.8) 21 (16.2)	0.891
Aboriginal or Torres Strait Islander status, *n* (%)	5 (5.6)	1 (0.8)	0.031
DKA, *n* (%)	54 (60.7)	56 (42.7)	0.009
Multiple DKA, *n* (%)	25 (28.4)	21 (16.0)	0.031
Hypoglycaemia admissions, *n* (%)	12 (13.5)	12 (9.2)	0.313
Multiple hypoglycaemia events, *n* (%)	4 (4.5)	2 (1.5)	0.185
Neuropathy, *n* (%)	8 (9.0)	9 (6.9)	0.564
Nephropathy ^†^, *n* (%)	6 (6.7)	4 (3.1)	0.197
Retinopathy, *n* (%)	8 (9.0)	3 (2.3)	0.025
Any long-term complication, *n* (%)	20 (22.5)	15 (11.5)	0.028
Hypertension, *n* (%)	32 (36)	45 (34.6)	0.838
Smoking, *n* (%)	27 (30.3)	17 (13.1)	0.005
Alcohol, *n* (%)	41 (46.1)	67 (51.5)	0.460
Driving requirement compliance, *n* (%)	33 (37.1)	56 (42.1)	0.071
HbA1c at entry, % (mmol/mol)	10.2 ± 2.6 (89 ± 29)	9.6 ± 2.6 (82 ± 29)	0.083
Last HbA1c, % (mmol/mol)	9.4 ± 2.4 (79 ± 27)	8.7 ± 2.2 (71 ± 24)	0.027
Target HDL, *n* (%)	41 (46.1)	75 (57.3)	0.046
Age at diagnosis, years	11.3 ± 5.8	12.1 ± 5.9	0.060
Diabetes duration, years	10.7 ± 6.4	9.1 ± 6.2	0.056
Number of clinic check ins	5.6 ± 4.0	6.6 ± 5.3	0.229
Number of years in clinic	1.8 ± 1.5	2.1 ± 2.0	0.389

Abbreviations: HbA1c = Haemoglobin A1c; DKA = diabetic ketoacidosis ^†^ > 2.5 mg/mmol for males and > 3.5 mg/mmol for females. *p* values were Chi-square test for discrete variables and *t*-test for continuous variable.

**Table 4 ijerph-18-12667-t004:** Unadjusted (OR) and adjusted odd ratios (aOR: 95%confidence intervals) of factors associated with mental health condition among young adults attending the diabetes transition clinic. Bolded CIs are significant associations.

	Model 1			Model 2		
Variables	OR	aOR [95%CI]	*p*-value	OR	aOR [95%CI]	*p*-value
Gender (1)	0.185	1.20 [0.67, 2.15]	0.533	0.162	1.18 [0.66, 2.08]	0.58
Type of Diabetes (1)	0.386	1.47 [0.56, 3.87]	0.434	0.389	1.48 [0.57, 3.82]	0.423
Age (/year)	−0.011	0.99 [0.89, 1.10]	0.832	−0.013	0.99 [0.89, 1.10]	0.808
Diabetes Duration (/year)	0.037	1.04 [0.98, 1.10]	0.194	0.032	1.03[0.98, 1.09]	0.249
HbA1c (1)	0.704	2.02 [1.10, 3.70]	0.023	-	-	-
DKA (1)	0.592	1.81 [0.93, 3.50]	0.079	0.764	2.15 [1.13, 4.07]	0.019
Hypoglycaemia (1)	0.212	1.24 [0.48, 3.19]	0.662	0.131	1.14 [0.45, 2.92]	0.784
Any microvascular complication (1)	0.487	1.63 [0.72, 3.66]	0.239	0.702	2.02 [0.92, 4.43]	0.08

DKA = Diabetic ketoacidosis; microvascular complications included retinopathy, nephropathy, neuropathy. The dependent variable was presence (1) or absence (0) of mental health condition and the independent variables included: Type of diabetes (type 1 and type 2 diabetes), HbA1c (≤9% (≤75 mmol/mol) and >9.0% (>75 mmol/mol)), presence/absence of at least one episode of DKA and hypoglycaemia, any long-term complication (Yes/No), and gender (male and female).

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
