# Peer review of "Eight-Year Retrospective Study of Young Adults in a Diabetes Transition Clinic"

_ijerph, 2021, doi:10.3390/ijerph182312667_

Round 1

Reviewer 1 Report

Thank you for an interesting study. On line #135 there is a reference made to "Figure", but there is no number by that word. Table 1 is lengthy, it takes up almost two pages. Perhaps dividing it into two tables might add to the ease of reading. On line 191 there is a reference made to "Figure 2", but I was unable to find Figure 2 in the manuscript.

Finally, I found that there is a statement on lines 198 and 199 that could be portrayed as biased against a group of people. The information indicated that "...mental health conditions were common in this population, particular" this should be particularly "...among Aboriginal people".  The problem is that there were only six (6) Aboriginal people in the study compared to 205 people born in Australia: according to Table 1 and on line 149 in the manuscript. A person could read the article and see that 83.3% (line 149) of Aboriginal people have a mental health issue and overlook the fact that there were only six (6) Aboriginals in the study.

Author Response

Reviewer 1

Comments and Suggestions for Authors

Thank you for an interesting study. On line #135 there is a reference made to "Figure", but there is no number by that word.

Response’; Number has been assigned to the figure. It now reads Figure 1

Table 1 is lengthy, it takes up almost two pages. Perhaps dividing it into two tables might add to the ease of reading.

Response: Table 1 has been split into two tables. We have also described the new table in the text. The added section now reads:

“Table 2 presents the clinical profile of the young adults, their treatment modalities and the diabetes-related complications reported in the transition clinic during the study period. Participants were mostly overweight, about one-third of them had hypertension and 21.1% had experienced multiple episodes of DKA over the study period. Insulin only treatment was predominant in this population and there was a reasonable proportion (n=18, 50.0%) that were on Oral/GLP1 treatment only”.

Numbering was revised for the rest of the table to match added table 2.

On line 191 there is a reference made to "Figure 2", but I was unable to find Figure 2 in the manuscript.

Response: Figure 2 has been provided.

Finally, I found that there is a statement on lines 198 and 199 that could be portrayed as biased against a group of people. The information indicated that "...mental health conditions were common in this population, particular" this should be particularly "...among Aboriginal people".  The problem is that there were only six (6) Aboriginal people in the study compared to 205 people born in Australia: according to Table 1 and on line 149 in the manuscript. A person could read the article and see that 83.3% (line 149) of Aboriginal people have a mental health issue and overlook the fact that there were only six (6) Aboriginals in the study.

Response: Thanks for this. We agree with the reviewer and revised the two sentences to put them in context of the small number of participants. In the first case in results, we revised the text to read:

“There were six Aboriginal people in this study, five of whom (83.3%) had a mental health condition”.

In the second case, we deleted the reference to Aboriginals here. It now reads:

“We have shown that mental health conditions were common in this population”.

Reviewer 2 Report

Thank you for giving me the opportunity to review this interesting paper.
A few notes follow:

Pg 2 line 60: HbA1c please write in full for the fist time

Pg 2 line 76: could you  explain who they are diabetes educators?

Pg 2 line 79 : The word "review" may confuse the reader. I suggest changing the word.

Pg 2 line 90: "The targets were set as" do you mean inclusion criteria?

I suggest to include in the conclusions what strategies you will use to improve the satisfaction of patients with mental illnesses, which seemed to be the population most at risk.

Author Response

Reviewer 2

Comments and Suggestions for Authors

Thank you for giving me the opportunity to review this interesting paper.
A few notes follow:

Pg 2 line 60: HbA1c please write in full for the fist time

Response. HbA1c has been used in full in the abstract and where it first occurred in the introduction. Subsequently abbreviation was used.

Pg 2 line 76: could you  explain who they are diabetes educators?

Response: We have explained who the Diabetes educators are in the methods. The section now reads:  Diabetes educators are clinicians who specialise in the provision of diabetes self-management education for people with diabetes.

Pg 2 line 79 : The word "review" may confuse the reader. I suggest changing the word.

Response: Audit has been used across the manuscript

Pg 2 line 90: "The targets were set as" do you mean inclusion criteria?

Response. No. This does not mean inclusion criteria rather it is the target level for optimal control of the clinical measures which are set by clinicians. The sentence was revised for clarity and now reads:

“The target levels for optimal control of the clinical measures were set as: …..”

I suggest to include in the conclusions what strategies you will use to improve the satisfaction of patients with mental illnesses, which seemed to be the population most at risk.

Response: Thank you for the comment. We have now expanded the section in the conclusion to highlight some of the strategies. The section now reads:

“Despite emerging strategies to help young adults with mental health issues, a diagnosis of a chronic disorder is an additional burden. Peer groups for affected individuals to interact with each other improves outcome of people with mental health condition[27]. Furthermore, having an in-house psychologist will initiate early intervention to improve patient satisfaction and optimal diabetes care. Strategies that involve the effective collaboration of primary care providers and mental health specialists/psychologists using principles of measurement-based stepped care and treatment to target the high risk population identified in this study can substantially improve patients’ health and functioning while reducing overall health care costs.”